# Who Did Spanish Politicians Start Following on Twitter? Homophilic Tendencies among the Political Elite

**Verónica Israel-Turim *** **, Josep Lluís Micó-Sanz and Miriam Diez Bosch**

Blanquerna School of Communications and International Relations, Ramon Llull University,
08022 Barcelona, Spain; joseplluisms@blanquerna.url.edu (J.L.M.-S.); miriamdb@blanquerna.url.edu (M.D.B.)
* Correspondence: veronicait@blanquerna.url.edu

**Abstract:** Political communication has undergone transformations since the advent of digital networks, but do these new platforms promote interactivity and a public sphere with a more democratic political debate or do they function as echo chambers of the elites? In this research, we study the accounts that Spanish politicians started following on Twitter from 2017 to 2020, with the aim of understanding whether they reproduce patterns of homophilic tendencies or if they give space to new voices. To do so, we selected a sample from the deputies that were in the Spanish parliament during the four years of the study and through a big data and machine learning software, we identified the accounts they started following as a network and categorized them. We combined manual and computational data analysis methods and used data visualization techniques to look for patterns and trends. The results suggest that the Spanish political elites exhibit homophilic behaviors in terms of account types and geographic proximity and present a gender balance among the accounts. This study also suggests that the behavior of the political elite presented particularities during the electoral period, where we can observe an intensification of the homophilic patterns.

**Keywords:** political communication; Twitter; homophily; social network analysis; social media; power elites; data visualization; echo chambers; digital communication; digital social networks

## 1. Introduction

### 1.1. Echo Chambers or Enhanced Public Sphere?

The way in which political communication is understood has changed since the advent of digital social networks (Alonso-Muñoz et al. 2016). These platforms have impacted the ways in which people interact, setting new dynamics of influence among members of power elites and in relation to the citizenry (Chadwick 2017; Jenkins 2008; Wallace 2018). Previous studies have pursued the objective of understanding if digital social media support the development of a diverse and inclusive public sphere where democratic discussion is promoted (Ausserhofer and Maireder 2013; Colleoni et al. 2014), given that they operate as an impulse for political activism (Feenstra and Casero-Ripollés 2014), habilitating new political actors and voices in the conversation (McGregor and Mourão 2016). Likewise, many authors claim that the digital realm helps the promotion of transparency and interac- tivity (Deuze 2011; Feenstra and Casero-Ripollés 2014; Shirky 2008), eliminating physical barriers (Ausserhofer and Maireder 2013) and traditional political and media gatekeeping filters (McCombs and Shaw 1972; Meraz 2009; Vargo 2018).

However, further studies show that instead of promoting such democratic participa- tion, in the digital sphere people strengthen their prior points of view (Ausserhofer and Maireder 2013) as they see the contents of those who they choose to follow, due to algorith- mically recommended content, which also tends to be in line with their views and opinions as they are based on search history and users' past activity (Finn 2017; Mayer-Schönberger and Cukier 2013; Terren and Borge 2021). This has led authors to speak about the internet as a space that deepens filter bubbles (Pariser 2011) and political polarization (Kubin and

von Sikorski 2021; Terren and Borge 2021). The platforms can mimic the capitalist dynamic of stratified attention, amplifying the messages of those who hold power (Casero-Ripollés 2021; Dubois and Gaffney 2014; Fuchs 2017), and previous studies show that the main recipients of politicians´ messages on social media are either politicians or the media, homophily being one of the reasons why they have been conceptualized as echo chambers of the elites (Bruns and Highfield 2013; Colleoni et al. 2014).

### 1.2. Homophily

"Similarity breeds connection" (McPherson et al. 2001, p. 415). The principle of homophily suggests that connections between similar people happen at higher rates than connection between people that present differences (McPherson et al. 2001), and that people tend to connect and create relationships with those who present similar characteristics to their own (Christakis and Fowler 2009; Katz et al. 2004; Kossinets and Watts 2009; Lauw et al. 2010; Lazarsfeld and Merton 1954; McPherson and Smith-Lovin 1987; McPherson et al. 2001; Perl et al. 2015). Moreover, people tend to strengthen their opinions by reading contents and following users aligned with their preexisting beliefs, instead of contacting with new or different perspectives (Christakis and Fowler 2009; Huber and Malhotra 2017; Katz et al. 2004; Lazarsfeld and Merton 1954; McPherson et al. 2001; Perl et al. 2015; Valera-Orda et al. 2018). When the principle of homophily is followed by the elites on social media, it can lead to the creation of echo chambers where the messages of those who already have power are amplified, gaining even more power (Bruns and Highfield 2013). It has been found that members of the elites such as politicians and journalists tend to follow and interact almost exclusively with other politicians and journalists (Bruns and Highfield 2013). In this framework, we wondered what is the case of Spanish politicians on Twitter. Do they interact with each other, or do they give space to the citizenship?

There is no consensus when referring to the concept of the political elite (Zuckerman 1977). There are different and complementary definitions of the concept, such as an elite that has a preeminent political influence (Roberts 1971); the Weberian model of elite power understood in terms of those who are in stable positions at the top of relevant social institutions (Wedel 2017); the concept of the elites as those who are in the position to make decisions that impact other individuals´ lives by being in the most relevant social hierarchies and institutions (Mills 1956); or as the minority that rules the society (Rahman Khan 2012). Moreover, elites can be understood under Meisel´s umbrella of the 3Cs, where there is group consciousness, coherence and conspiracy among the members of a power group (Korom and Planck 2015; Meisel 1958; Zuckerman 1977). Therefore, in the present research we studied the Spanish political elite from the perspective of a power group that exercises high influence and can be analyzed as a cluster, as it represents those who were in a hierarchical position in one of the most influential institutions, the parliament, enabling them to make decisions that affect the rest of the members of the society. They were the deputies who constituted the parliament from 2017 to 2020, analyzing only those who shared the entire period, with the purpose of generating a first approximation to their behavior regarding the type of accounts they began following as an elite. They were heterogeneous in terms of party affiliation, gender, age, origin, among other variables, but homogeneous in terms of the social role they occupied in the studied period, and therefore homophily can be measured in terms of similarity to the determined sample. We believe there are lines to further explore in future research by subcategorizing this elite in different periods, by political party or by gender. In the present research we studied the Spanish political elite as a group, taking into account the positional method of elite studies (Best and Higley 2017; Hoffmann-Lange 1989) that states that political power and influence in societies is conferred by formal institutional positions in the main organizations where decisions that affect the citizenship are taken, as well as the institutions responsible for the resources' social distribution (Best and Higley 2017). The elite structure is pluralistic, nonetheless "theorists acknowledge that modern democracies are organizationally diverse, they claim that the diversity of organizations and interests they embody are not reflected in

the elite structure. They assume that power is more concentrated in a small power elite than exponents of pluralism believe, so that participation in crucial policy decisions is limited to a small circle or knot of actors with common social backgrounds and interests that are concealed by a diversity of organizations and interests that, in terms of decisive power, is more apparent than real" (Best and Higley 2017, p. 80).

Homophily can be driven by different dimensions, such as geographical position, race, ethnicity, religion, sex, gender, age, network position, and beliefs, among other things (Lazarsfeld and Merton 1954; McPherson et al. 2001). In this research we focused on analyzing whether the Spanish deputies started to follow mostly political and media accounts, or if they started to follow citizenship accounts, taking into account the tendency that politicians and media have shown to follow and interact with each other, as found in previous research (Ausserhofer and Maireder 2013; Bruns and Highfield 2013; McGregor and Molyneux 2018; Molyneux 2015). We also studied the location of the accounts they started following, as the geographical position is a well stated form of homophily found to be reproduced also in online connections (Ausserhofer and Maireder 2013; Casero-Ripollés 2021). We also focused on understanding if the accounts they started to follow presented a balance between women and men, since we found an exhaustive amount of previous research that shows the long patterns of misrepresentations of women in political elites and power positions in general (Aaldering and Van Der Pas 2018; Bode 2016; Carli and Eagly 2002; Connell 1987; Kubu 2017; Lombardo 2008; Lovenduski 2005; Madsen and Andrade 2018; Painter-Morland 2011), and even when being in powerful positions, they can remain as outcasts of the inner circles of the elites (Moore 1988). Moreover, even when having balanced gender representation, an equal number of women representatives in the government does not necessarily mean that there will be a qualitative representation of women's interests (Lombardo 2008). Regarding social media interactions, it has been stated in previous research how male journalists and politicians tend to interact with a majority of male peers (Colleoni et al. 2014; Usher 2018), whereas such inbred homophily has not been found among women journalists (Maares et al. 2021). Given the persistent evidence of off- and online gender inequalities in politics, this research also seeks to examine how gender dynamics impact the way Spanish politicians relate to each other regarding the accounts that the Spanish parliamentarians start following on Twitter.

### 1.3. Twitter, the Political Network?

"Twitter is the de facto social media platform for discussing politics online" (Chamberlain et al. 2021). Twitter has been described as a political tool (Pérez-Curiel and Limón Naharro 2019; Redek and Godnov 2018) and as a political network (Conway and Wang 2015; Fernández Gómez et al. 2018) as it represents a significant role in political communication campaigns (Alonso-Muñoz et al. 2016; Usher 2018). Previous research shows that it is one of the social platforms preferred by politicians and political parties (Alonso-Muñoz et al. 2016). More than 80% of opinion leaders are on Twitter (González Bengoechea et al. 2019; Smith 2020), and in Spain, previous research has found that more than 90% of the deputies are users of this platform (Haman and Školník 2021). Political actors use this platform to broadcast their messages and for political debate, as well as to interact with opinion leaders and key actors (Ausserhofer and Maireder 2013; Broersma and Graham 2013). Nonetheless, as mentioned above, this interaction tends to be with other politicians and journalists, not with the citizenship (Alonso-Muñoz et al. 2016; Cervi and Roca 2017).

Twitter research has become very popular in the past few years as Twitter provides access to large amounts of available digital data (Williams et al. 2013; Zimmer and Proferes 2014). Previous literature states that most Twitter studies focus on content analysis (Zimmer and Proferes 2014). Twitter research on echo chambers has focused on interactions and content exposure, and the methods can vary, using digital trace data and self-reported data (Terren and Borge 2021). Political communication has been approached in Twitter studies in different research areas such as the use of the platform in determined events, its use

by the public, and the use that political parties and politicians make of the microblogging network (Chamberlain et al. 2021; Jungherr 2016).

In Spain, Twitter research has focused on the identification of influential actors in the political conversation using big data to detect digital authority (Casero-Ripollés 2021), and the use that Spanish political leaders make of the social platform analyzed from different perspectives such as in comparison to politicians from different political systems such as the United Stated of America and Norway (Cervi and Roca 2017), to detect the influence degree and the types of strategic communications tactics that the Spanish leaders use on Twitter, as well as analyzing the interconnection between the politicians' Twitter and media profiles (Suau-Gomila et al. 2020), or regarding the linguistic strategies that politicians use in self-referencing (Coesemans and De Cock 2017). Moreover, previous research on Twitter in Spain has focused on gender gaps among politicians, showing how there are still differences between the attention and amplification that women receive in the political Twitter sphere (Guerrero-Solé and Perales-García 2021), the differences in the language used between men and women politicians (Beltran et al. 2021), as well as the differences between women and men politicians from different Spanish parties when tweeting about feminist issues (Fernández-Rovira and Villegas-Simón 2019).

In this research we focused on analyzing the accounts that Spanish politicians began following, with the aim of contributing to the research on the use that political actors make of Twitter in Spain from a gender perspective, which even though has been previously explored (Beltran et al. 2021; Casero-Ripollés 2021; Cervi and Roca 2017; Coesemans and De Cock 2017; Fernández-Rovira and Villegas-Simón 2019; Jungherr 2016; Stier et al. 2018; Suau-Gomila et al. 2020), still lacks the consideration of homophily among Spanish political elites on Twitter. Moreover, research on following flows on Twitter in Spain among politicians is practically non-existent.

*1.4. Followership*

Why are we analyzing who the politicians follow? On the one hand, the accounts users follow on social networks determine their experience on that network by defining the content to which they are exposed. Earlier studies show that the content users see on their social media feeds influences their perception of the relevance of these topics (Feezell 2018) but also, depending on the accounts they follow, the algorithmic recommendations they receive from the network (Gupta et al. 2013; Hutchinson 2017; Twitter 2019b). One of the criteria used by Twitter´s algorithm to create recommendations is to suggest the accounts followed by the accounts each user follows (Twitter 2019b), which means that the accounts followed by relevant users and influencers usually gain more visibility on digital platforms as they tend to be more algorithmically recommended to other users (Twitter 2019a). Therefore, the accounts that the Spanish deputies follow may be recommended more frequently to the users that follow them, gaining more visibility, influencing the whole network.

## 2. Materials and Methods

With the aim of understanding the behavior of the Spanish politicians regarding who they started following on Twitter, we created a sample of deputies. This sample was composed by the deputies that coincided in the parliament during the studied period, which covered the years 2017 to 2020. To define the sample, we made a database with all deputies who made up the parliament between 2017 and 2020 and then proceeded to select those who coincided during these four years. This means that all those deputies who were only there during a shorter period within those years, and not the whole period, were removed. This way, we were left with those who shared the four years of parliamentary duty.

We manually checked the number of followers, location and gender of the members of the sample and once we identified them, we proceeded to create a network, understood as such according to social network analysis (Barnes and Harary 1983; Casero-Ripollés 2021; Grandjean 2016; Tang and Liu 2010), in order to analyze them. We used a machine learning

software named *Contexto.io*, which was developed as part of the project "Influencers in Political Communication in Spain. Analysis of the Relationships Between Opinion Leaders 2.0, Media, Parties, Institutions, and Audiences in the Digital Environment". This software can organize, explore and analyze contexts of information around people using their public digital footprints. A context is composed by a group of people and/or organizations that interact forming an ecosystem. They are created by using their Twitter accounts which are then algorithmically sorted by their relevance within the context, taking into account their digital trace. Therefore, we performed a manual search of each of the deputies on Twitter to identify their user accounts. Utilizing the abovementioned software we created a new group and manually added each Twitter user and thus created the network with the 97 Twitter accounts of the deputies who coincided in the Spanish parliament between 2017 and 2020. Once the network was created, this software organized the accounts in a graph regarding different possible parameters such as relations, communication, common organizations and predicted links, which are the categorizations we selected for the present sample. The resulting network, composed of 97 deputies, 54 men and 43 women, is the following (Figure 1):

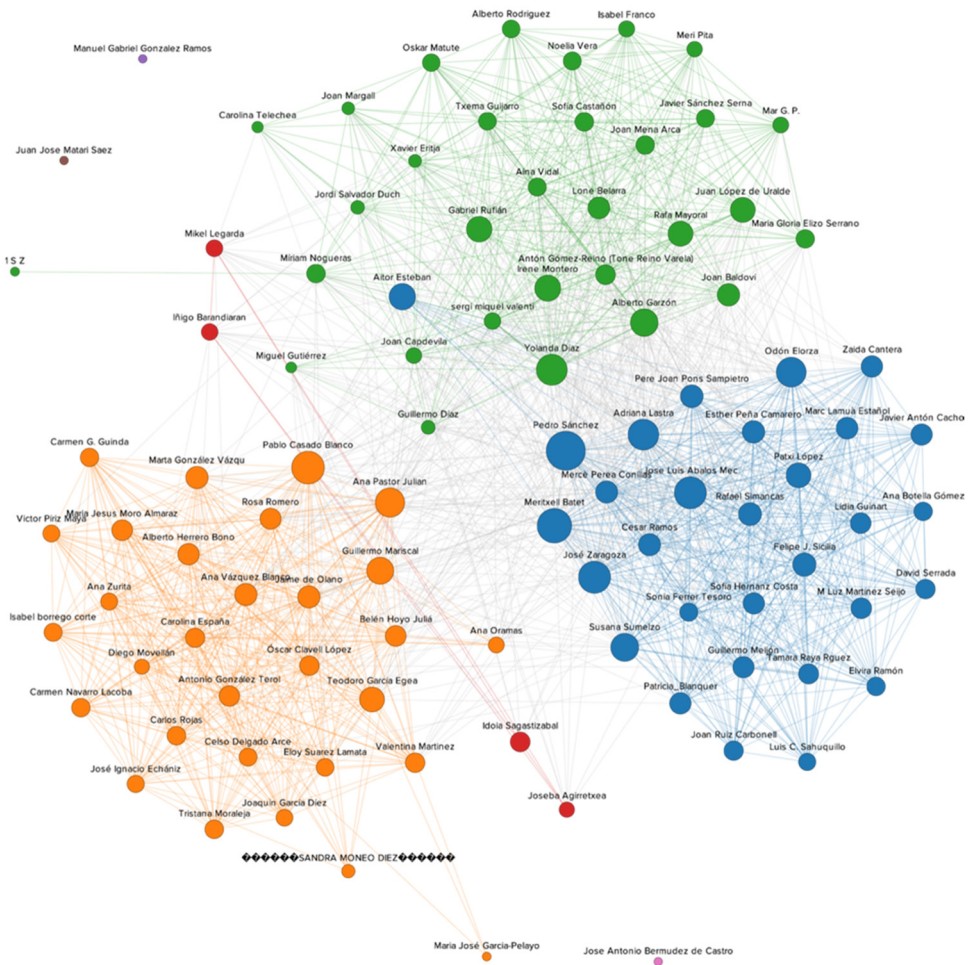

**Figure 1.** Network of Spanish deputies sample graph.

Once we created the sample, we consulted the data regarding who they started to follow in different periods. The sample, composed of all the deputies that coincided in the Spanish parliament from 2017 to 2020, is understood as one possible group to define the stable political elite of those years, in order to have a sample with sufficient members to analyze as a conjunct. We could have categorized the sample in many ways, taking into account the politicians' gender, race, origin, political affiliation, religious affiliation, and

analyzed homophilic tendencies from these possible different categories (McPherson et al. 2001). The present study represents a specific case study on Spanish politicians on Twitter, so we decided to make an approximation to the homophilic behaviors of the whole political class that composed the parliament during four years, making an approximation to the macro category as politicians in power, to see if they started to follow the citizenry or if they started to follow mainly other politicians and media, as stated in previous research on echo chambers and homophily on Twitter (Bruns and Highfield 2013; Colleoni et al. 2014). Methodologically, in elite studies, there are three main ways of determining an elite for its study: positional, decisional and reputational (Best and Higley 2017; Hoffmann-Lange 1989), also categorized as reputational, structural and the agency or decision-making approach (Scott 1974). In the present study, we have taken the positional/structural path, since, as Scott states: "the structural approach has the most to offer to researchers on power and that it provides a basis for incorporating the insights of the rival approaches" (Scott 1974, p. 84). Taking into account theoretical and pragmatic reasons, the positional method is one of the most widely used in the study of national elites (Best and Higley 2017; Hoffmann-Lange 1989; Larsen and Ellersgaard 2017, p. 53). Given that the present study is a first approach to the political homophilic tendencies regarding the accounts that the Spanish political elite began following, we believe that the best methodological approach is to select the sample according to its formal position of power in society, in this case the set of deputies that form the Spanish parliament. Structural approaches to power are centered on the aspects of strategic positions in the main institutions of a society; positions that are the at the core of the resource's distribution and control, which are the main centers of power, and therefore, those who occupy these positions are understood as main actors in the exercise of power. Therefore, the sample clearly represents an elite and seeks to provide an approximation of the political elite in Spain. Like any method and methodological decision, it has advantages and disadvantages. The advantage in this case is to be able to understand how the Spanish elite operates as a whole, as a group of decision-makers, as a cluster of people with positions of high impact on citizens' lives. The limitation of this approach is to leave aside the differences among them, such as gender, political orientation, nationality and the language they speak. We believe it would be interesting to deepen into the abovementioned subcategories in future research, subsequently to the present project that aims to analyze the parliamentary Spanish elite as a group, as even though they are heterogenous, the political elite´s diversity has been presented by authors as more apparent than real, taking into account that they share involvement in central policy decisions (Best and Higley 2017). Moreover, we followed the methodological approach of several previous studies where the political elite was analyzed as such, leaving aside the differences among them, such as their political affiliation or gender (D'heer and Verdegem 2014; Putnam 1976; Sjöberg and Drottz-Sjöberg 2008; Verweij 2012).

We were also able to access the data of the accounts they started following through the *Contexto.io* software, which has a section called *Expand* where it is possible to visualize the accounts that the context started to follow, with possibility of selecting specific periods to analyze. This section provides the option to select whether to display the accounts that the group started to follow including those belonging to the context or excluding them or to display only those that were outsiders of the network. The software thus provides a list in order of popularity within the network, measured by the percentage of users in the group that started following each account. For this study, we chose to visualize the accounts that the sample started to follow both, in-network and out-of-network. We studied the 50 accounts that the sample began to follow in highest percentages in 2017, 2018, 2019 and 2020. We considered 50 or more accounts generated a high dispersion. These accounts were manually catalogued in order to proceed to search for patterns and trends (Batrinca and Treleaven 2015; Dodge 2005; Mahrt and Scharkow 2013; Vogt et al. 2014) that could help us understand the relationships and influence flows of the analyzed politicians and other groups such as the media and the citizenship, and to be able to comprehend the space

women have in the politicians cybersphere. The categories used to analyze the accounts the sample started to follow were the following.

### 2.1. Type of Account: Political, Media or Citizenship

The political accounts were sub-categorized in political parties, politicians and public institutions. Public institutions are included in this category as they can be considered political devices that may operate according to the political framework (Thoenig 2003). The media accounts were divided into media institutions and journalists. The citizenship accounts were classified as civil institutions (constituted by NGOs, civil organizations, companies, entrepreneurships, etc.) and users (including scholars, entrepreneurs, influencers, celebrities, artists, activists, etc.).

### 2.2. Person/Institution

We categorized the accounts considering whether they belonged to a person or an institution.

### 2.3. Location

The location is the place or precedence of the accounts the deputies started following expressed in their Twitter user accounts.

The data we analyzed in this research corresponded to the accounts that the sample started following between 2017 and 2020, not the set of accounts followed by the network, since it is not possible to access this data, taking into account that users start following and unfollow accounts dynamically.

### 2.4. Number of Followers

The number of followers of the accounts was categorized in five levels defined in previous research (Table 1):

**Table 1.** Number of followers categorization.

| Influencer Category | Number of Followers |
| --- | --- |
| Non-Influencers | <1000 |
| Micro-Influencers | 1001–10,000 |
| Mid-Influencers | 10,001–100,000 |
| Macro-Influencers | 100,001–1,000,000 |
| Icon-Influencers | >1,000,000 |

Source: (Israel-Turim et al. 2021).

The number of followers used in the analysis corresponds to the period in which the study was being carried out, not to the number of followers the accounts had when the sample started following them, as we cannot access this data.

### 2.5. Gender

From the accounts that belonged to people we categorized them according to the gender they identified themselves with by analyzing their profiles. To do this, we took into account how they described themselves in their bios and if their bios did not make it clear, we looked for more information online about each user to find out how they defined themselves. Since most of them used Spanish and Catalan, which are languages that contain gender differentiation in most of the words, it was easier to identify how they referred to themselves, since by putting for example "deputy" in their bios, which would be "diputada" or "diputado" or "diputade" in Spanish, we can already know how they identify gender-wise, as "a" is used for women, "o" for men and "e" for non-binaries. Another example is an account whose bio was "Un socialista vasco", which translates as "A Basque socialist". This phrase in Spanish clarifies the gender the user identifies with, as the pronoun is masculine. The gender subcategories were women, non-binary and men

(Butler 1988; Richards et al. 2016), aiming to explore gender balance (or imbalance) trends, as women and dissidents have a long tradition of being underrepresented in powerful positions (Carli and Eagly 2002; Connell 1987; Kubu 2017; Madsen and Andrade 2018; Painter-Morland 2011). Previous research has shown a problematic confusion between sex and gender, which tend to be presented as interchangeable categories, when sex has been defined as a biological phenomenon whereas gender is understood as a cultural dimension (Bittner and Goodyear-Grant 2017). Both, sex and gender, tend to be understood as binary categories, male and female in the case of sex, and men and women in the case of gender, whereas research has proven that both are not. There is a percentage of the population that is born as intersex or third-sex (Carpenter 2018), estimated to be around 1.7% (Amnesty 2018), and there are other gender identities such as genderqueer and non-binary (Richards et al. 2016). In this study, following previous research where identities who do not identify themselves in a binary way as women or men are taken into account, we categorized the accounts into women, men and non-binary (Medeiros et al. 2020).

The analysis of political ideology is a limitation of the present research, in which we decided to focus on the types of accounts, number of followers, geographic location, and gender. We consider it is relevant to delve into more variables of analysis in future research, such as political ideology.

## 3. Results

### 3.1. Types of Accounts

The Spanish deputies that coincided in the parliament in the four years of this study started to follow a majority of political accounts, with more than 50% every year, presenting a homophilic behavior regarding the type of account they began to follow (Colleoni et al. 2014; McPherson et al. 2001) (see Figure 2).

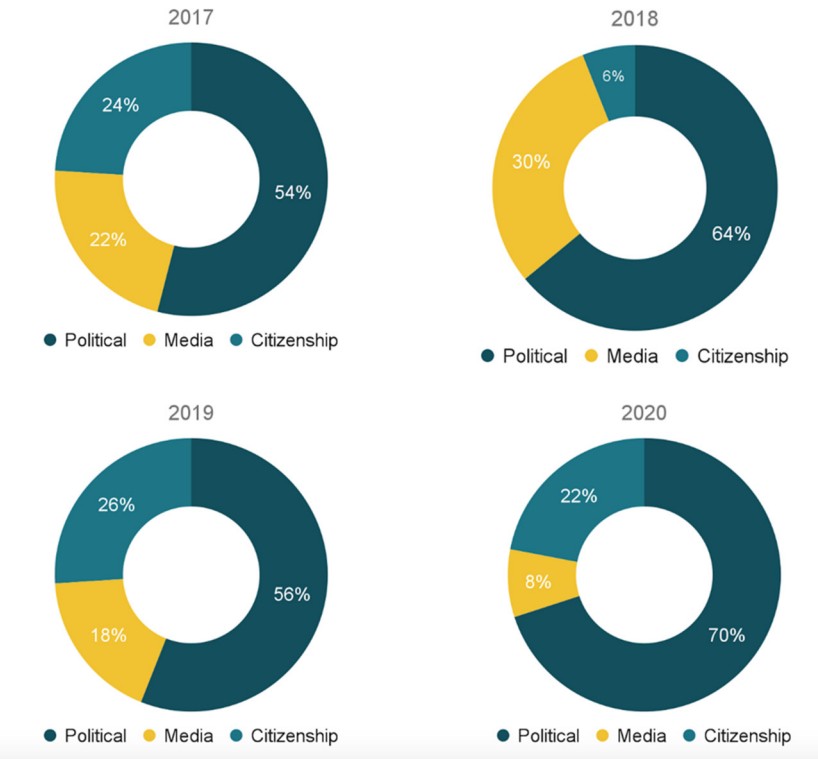

**Figure 2.** Percentages of the types of accounts the sample of Spanish deputies started to follow in 2017, 2018, 2019 and 2020.

The years in which we can find a higher percentage of political accounts were 2018, an electoral year in Spain, and 2020. During the electoral year, the media accounts that the

sample started to follow increased, being this the year in which they started following the highest percentage of media accounts, with a 30%. The rest of the years, the sample began following more citizenship accounts than media ones, though on average, they started to follow the exact same percentages of media and citizenship accounts. The year with the lowest percentage of media accounts was 2020, which was not a predictable result, as it was the year in which the COVID-19 pandemic began, and digital and social media consumption increased notably (Singh et al. 2020). The fact that they began following more than 20% of citizenship accounts every year, except in 2018, can be understood as a shy openness to listen to voices outside of the media and political elites, and may also be explained by the raise of the influencers figures, who are gaining relevance in the online sphere (Fernández Gómez et al. 2018; Pérez-Curiel and Limón Naharro 2019).

Political Subcategories

The vast majority of the political subcategories that the sample began following were other politicians. The year in which they began following fewer politicians was 2018, the electoral year in Spain, when the politicians accounts still represented 66% of the political accounts the sample began following. This year was the year in which they began following more public Institutions, which included several ministries, the Moncloa account and the European Parliament (Figure 3).

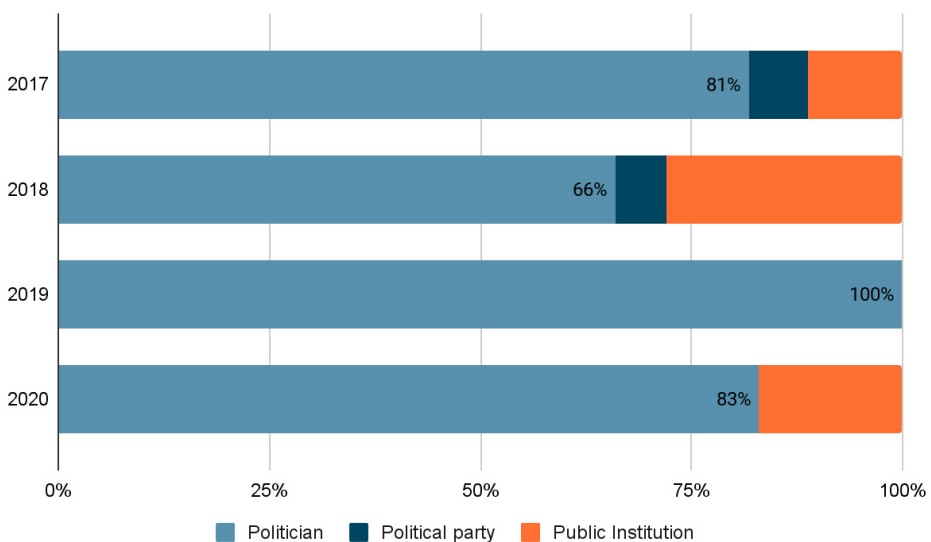

**Figure 3.** Percentages of political subcategories that the sample of Spanish deputies started to follow in 2017, 2018, 2019 and 2020.

The political parties' accounts were the subcategory less followed by the politicians' network. An explanation for this may be that there are fewer political parties than politicians, as there are many politicians per party. Another possible justification is that they already followed the political parties' accounts, or the fact that this network is constituted by deputies from different political parties, so they did not coincide in following them. We believe analyzing whether the politicians follow the accounts of the political parties that they do not belong to, and who follows each political party, constitutes an interesting line for future research.

*3.2. Institution or Person*

The percentages (Figure 4) of accounts that belonged to individuals and institutions were very similar to the percentages presented in the accounts of the political subcategories, which makes sense, since an average of 60% of the accounts that they started to follow were political. The tendency of Spanish politicians is to follow accounts belonging to individuals as opposed to institutional accounts. The analyzed politicians seem to give more space to

people than to institutions among the accounts they started following on Twitter. From the institutional accounts they began following, the majority were political institutions (public institutions or political parties), media institutions in second place, and the civil institutions were the least followed. The year in which they started to follow more institutions was 2018, when they started following 32% institutional accounts, of which 69% were political institutions and 31% were media institutions. It was the only year in which they did not start to follow any civil organization (Figure 5).

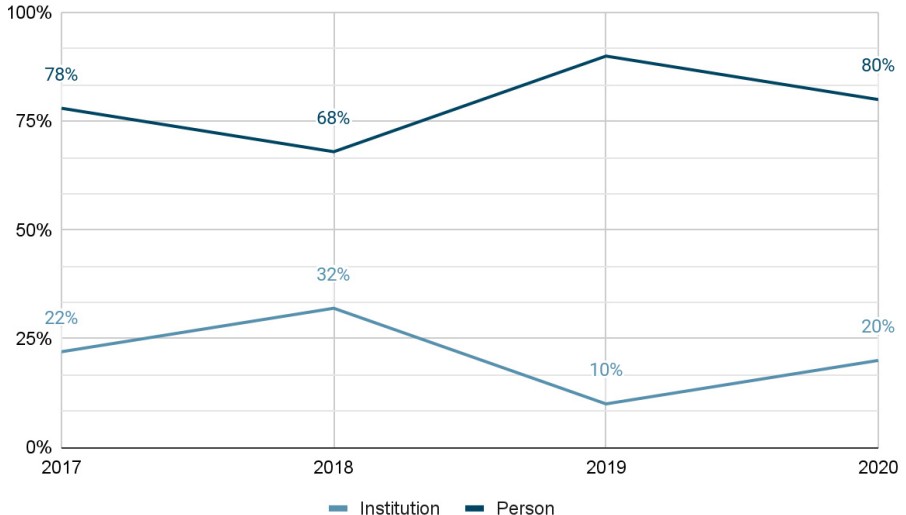

**Figure 4.** Institutions vs. people percentages that the sample of Spanish deputies started to follow in 2017, 2018, 2019 and 2020.

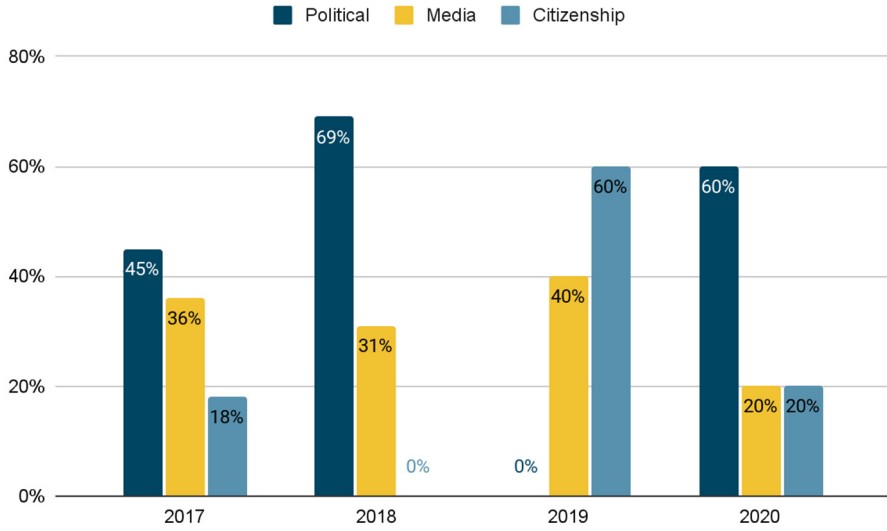

**Figure 5.** Percentages of the types of institutions that the sample of Spanish deputies started to follow in 2017, 2018, 2019 and 2020.

### 3.3. Location

Once again, the year 2018 presented differences in comparison to the rest of the years of the study, as during it the sample did not start to follow accounts from any country other than Spain. The rest of the years, only 4% of the accounts followed belonged to other countries. The countries from where the sample began following accounts were the United States of America, England, Sweden, and Belgium, countries that belong to the global north. We could not find any accounts from countries of the global south, defined as the countries

that tend to be marginalized in the political sphere (Medie and Kang 2018). This result also supports evidence of homophilic behavior (McPherson et al. 2001) (Figure 6).

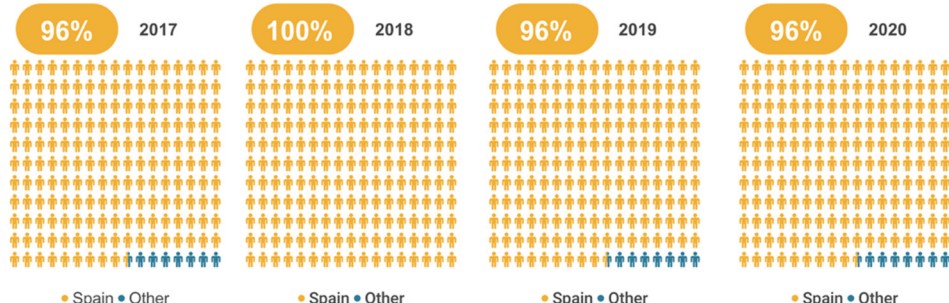

**Figure 6.** Percentages of Spanish accounts or those from other countries that the sample of Spanish deputies started to follow in 2017, 2018, 2019 and 2020.

*3.4. Number of Followers*

Most of the accounts that the analyzed Spanish deputies began following had between 10,001 and 100,000 followers, categorized as mid-influencers. This trend was especially high in 2018 and the pattern in all the years of the study, except in 2017, when we found almost the same number of micro- and mid-influencers, with one more account of micro-influencers.

The Spanish deputies began following a similar number of accounts from micro- and macro-influencers, with one more account belonging to the micro-influencers. In the fourth place, they began following icon-influencers and the non-influencers were the group least followed by the sample (Figure 7).

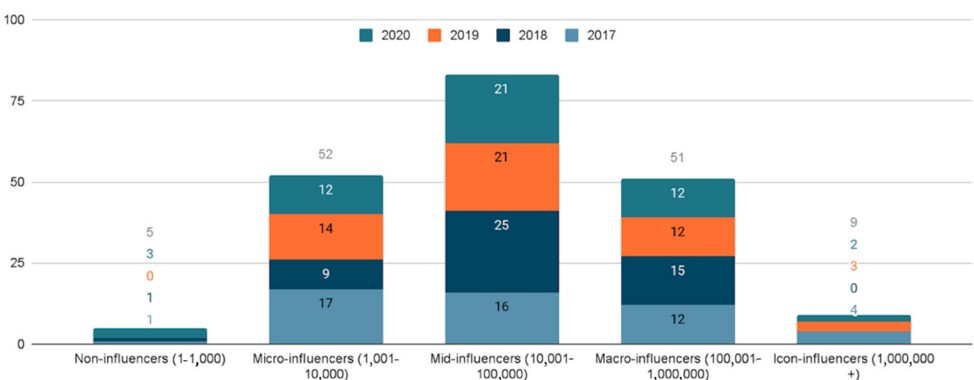

**Figure 7.** Aggregated number of followers of the accounts that the sample of Spanish deputies started to follow in 2017, 2018, 2019 and 2020.

In order to comprehend whether this result implies homophilic behavior, we analyzed the number of followers of the accounts in the sample.

As we can observe, the distribution of the number of followers in the sample is not the same as that of the accounts they started following. While the accounts they began following were a majority of mid-influencers in the first place and micro- and macro-influencers in very close second and third places, the sample was constituted by accounts that were mainly micro-influencers in the first place, mid-influencers in second and macro-influencers in third place. While this could be understood as a difference between the composition of the sample and the accounts they followed, and therefore non-homophilic behavior, most of the accounts in both networks remained split between micro-, mid- and macro-influencers. In any case, we can see that non-influencers and icon-influencers were the types of accounts that had the least presence. From this point of view, we can say that

the behavior of the sample was to follow accounts similar to their own in terms of number of followers (Figure 8).

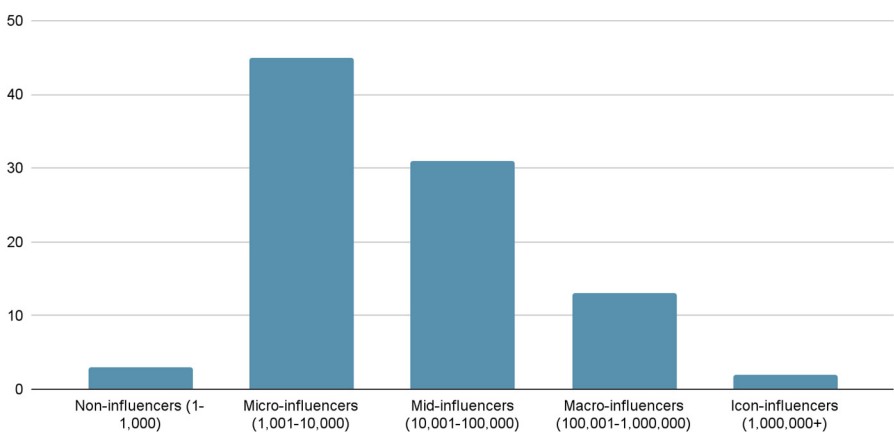

**Figure 8.** Sample accounts and numbers of followers of the accounts.

*3.5. Gender*

During the first three years of this study, the Spanish deputies started following more women than men; 2020 was the only year in which they began following more men than women. The Spanish senate as a whole was composed of 62% men and 38% women senators, and has presented a similar distribution for the past five legislatures (Senado 2020). The sample of the present study constituted 56% men and 44% women, which represents a more balanced network, especially considering the long underrepresentation of women in powerful positions (Carli and Eagly 2002; Connell 1987; Kubu 2017; Madsen and Andrade 2018; Painter-Morland 2011) (Figure 9).

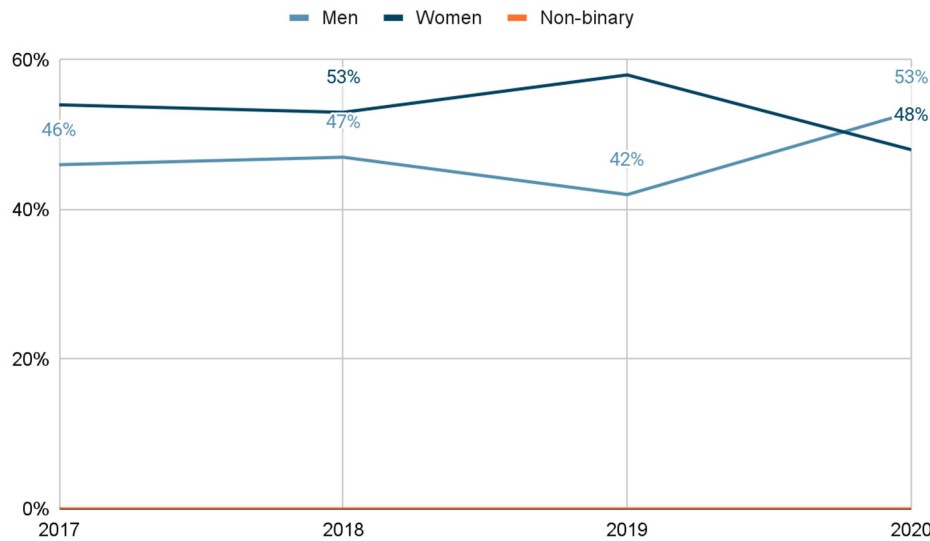

**Figure 9.** Gender of the accounts that the sample of Spanish deputies started to follow in 2017, 2018, 2019 and 2020.

When analyzing the gender per category, we can observe how the sample of analyzed Spanish politicians started following a higher percentage of female politicians (69%) than of male politicians (57%), with similar percentages of women and men in accounts belonging to journalists (17% and 16% correspondingly), and a higher percentage of male users (27%) over female users (14%). The user category included entrepreneurs, scholars, celebrities, athletes and activists. We wonder at the reason for following more male users than fe-

male ones. May it reflect a tendency to follow women only when they have a very clear established position, such as a political role? (Figure 10).

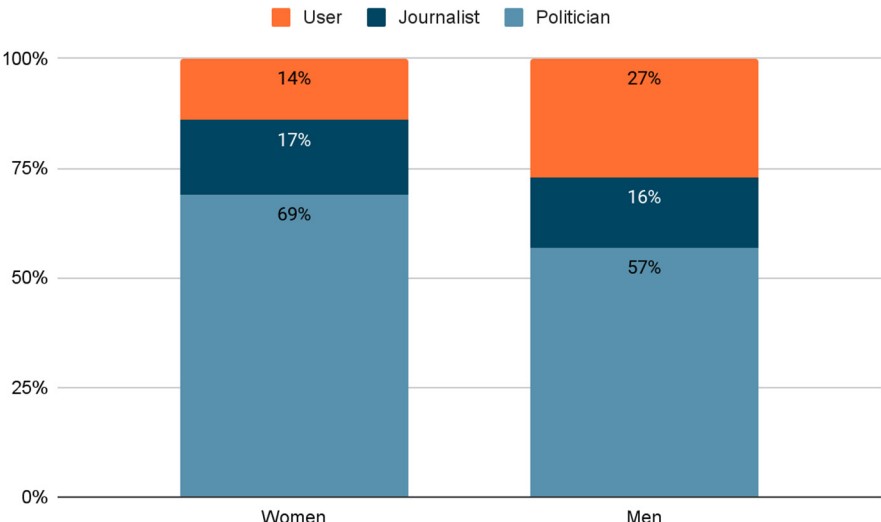

**Figure 10.** Gender per category of the accounts that the sample of Spanish deputies started to follow in 2017, 2018, 2019 and 2020.

## 4. Discussion

In this research we analyzed the accounts that the deputies that coincided in the Spanish parliament from 2017 to 2020 began following as a group, with the aim of searching for patterns and trends (Batrinca and Treleaven 2015) that could help us understand the influence flows between politicians, other power groups such as journalists and media, and citizens. Moreover, we sought to comprehend if they reproduce homophilic behavior on Twitter (McPherson et al. 2001) by starting to follow members of other power groups such as other politicians or the media, and therefore conceived it as an echo chamber of the elites (Bruns and Highfield 2013), or if they gave space to the citizenry, promoting a democratic and inclusive political debate and public sphere (Ausserhofer and Maireder 2013; Colleoni et al. 2014).

The analyzed Spanish deputies, who corresponded to the ones that coincided in the parliament between 2017 and 2020, started following a majority of political accounts. More than half of the accounts they began following every year were political and among these, the majority were of other politicians. Given the fact that choosing to follow accounts that presented the same characteristics as their own, in this case other politicians, which would constitute the dimension of others that share their own sociopolitical status, working sphere and role in the society (McPherson and Smith-Lovin 1987), we can consider that the present results provide evidence to support the theory of homophilic behavior among the political Spanish elite (Colleoni et al. 2014; McPherson et al. 2001), considering that the politicians started following mainly other politicians. Nonetheless, the pattern of following other power elites only applied to the political elite, as the sample did not begin to follow more media than citizenship accounts. In fact, on average, they began following the same percentages of media and citizenship accounts, though the distribution differed. During 2017, 2019 and 2020, the network of Spanish politicians began following more citizenship accounts than media ones. They began following between 22% and 26% citizen accounts, which may imply that part of the politician's attention goes to seeking views of the citizenship and interacting with them. This result is in line with studies that state that the figure of the influencer, which has emerged in the past few years (Fernández Gómez et al. 2018; Pérez-Curiel and Limón Naharro 2019), is making room for new voices in different areas, including the political sphere, redefining social influence towards a gradual redistribution of power (Casero-Ripollés 2021). However, in 2018 the sample only started

following 6% citizenship accounts, which means that during the electoral period is when Spanish deputies opted to start following fewer citizenship accounts. In this year, they began following 64% political accounts and 30% media accounts. This result is aligned with the studied link between politicians and journalists and their dialogical co-creation of the public and political agenda (Barberá et al. 2019; Davis 2007; Harder et al. 2017; Martin 2014; McCombs and Shaw 1972). Contrarily, 2020 was the year in which they began following the lowest percentage of media accounts, which seems unforeseen taking into account the fact that this year the COVID-19 pandemic began, and the media and information consumption increased considerably, media being considered a fundamental tool for the health emergency management (Casero-Ripollés 2020; Singh et al. 2020). We may reason that the politicians already followed the media accounts, so when the pandemic started, they already had the accounts among the ones they followed, which is why they did not start following them that year. However, further research would be needed to answer this matter, as one of the limitations of this study is that we analyzed the accounts they began following, as we could not access the data of the accounts they were already following. Another hypothesis for this result is that they accessed the pandemic information in a more direct way in the parliament, and therefore they did not need to follow media accounts for this purpose.

Regarding whether they started to follow institutions or individuals, the trend among the analyzed accounts was to follow fewer institutional accounts and more personal ones. Among the institutional accounts they followed, most were political (public institutions or political parties). In second place they followed media institutions, and the type of institutions they started following to a lesser extent were civil institutions. The year in which we can find more institutional accounts was 2018. It seems like the analyzed deputies preferred creating new connections with users like them, and during the electoral year they displayed a different behavior, following more institutional accounts. Moreover, 2018 was the only year when they did not start following any civil organization accounts.

Most of the accounts that the Spanish deputies started following were Spanish accounts, once again presenting homophilic behavior, this time concerning geographical proximity (Katz et al. 2004; McPherson et al. 2001). Moreover, the accounts they began following from other countries were all from the global north, which can also be understood as homophilic behavior and as the use of Twitter as an echo chamber of the elites (Bruns and Highfield 2013; Colleoni et al. 2014; Meraz 2009), given that it represents a perpetuation of the north−south global geopolitics hierarchy (Medie and Kang 2018), where "the voices representing the developing world are hardly heard" (Vu et al. 2020, p. 460).

Regarding the gender of the accounts that the Spanish deputies started following, we found that they began following more women than men during the first three years of the study, and in the fourth year of the study the difference was 5% more men. This result defies long patterns of misrepresentations of women in political elites and powerful positions in general (Aaldering and Van Der Pas 2018; Bode 2016; Carli and Eagly 2002; Connell 1987; Kubu 2017; Lombardo 2008; Lovenduski 2005; Madsen and Andrade 2018; Painter-Morland 2011). The fact that the sample constituted 56% men and 44% women may be one of the reasons for this result. Nonetheless, we believe it is important to further explore this issue, given that there may be other aspects that influence this outcome, such as the fact that perhaps the sample already followed male politicians and during the years of the study, from 2017 to 2020, the feminist movement in Spain gained relevance (Willem and Tortajada 2021), which may have influenced politicians to start following more women. It is also important to keep in mind that the constitution of the analyzed sample contains different political parties that may have had greater or lesser affiliation with feminist ideas. We consider that it would be relevant to study in future research whether this balanced percentage between men and women is maintained when studying each political party separately.

The sample started following similar percentages of women and men journalists (17 and 16%, respectively) but started following more women politicians than men politi-

cians (69 and 57%), and more men users than women users (27 and 14%). The category of users included businessmen, celebrities, influencers and academics, among others. Men are given space in various and different roles and are often taken as referents and leaders in different fields, as there is a long association of masculinity and leadership (Aaldering and Van Der Pas 2018). Evidence of this is the media´s gender bias in the use of a higher number of male sources in the most diverse areas, regardless of whether there are women leaders in the areas being consulted (Armstrong 2004; Armstrong and Gao 2011; Armstrong and Nelson 2005; Bustamante 1994; De Swert and Hooghe 2010; Moreno-Castro et al. 2019; Zoch and Van Slyke Turk 1998). The results of this study propose the idea that women begin to be followed when they have an established role such as a political office, and men are taken as referents in a wider variety of fields.

During the year 2018, which was an electoral year in Spain, we observe a few particularities. It is the year in which the sample began following more media accounts. This makes sense in an electoral context, as media and journalists are relevant actors of influence on political agendas (Davis 2007). This same year, they began following more political institutions within the political accounts, and it is the year with the highest percentage of institutions in general. Although the general trend concerning the location of the accounts was to start following a vast majority of Spanish accounts with more than the 90% every year, the only year in which there were no accounts from other countries was 2018. These results suggest that the electoral year impacted the behavior of the political elite on Twitter in relation to who they started following. Although the Spanish politicians analyzed showed homophilic behavior in terms of the accounts they began to follow during the entire period studied, we can observe an intensification during the electoral year, being the year in which they began to follow more media accounts, more institutional accounts, more public political institutions and more accounts from Spain, and one of the years in which they began to follow fewer women. Therefore, we can conclude that the Spanish deputies showed homophilic behavior during the period from 2017 to 2020 regarding the accounts they started to follow in terms of type of accounts (political, media or citizenship) and the gender, the number of followers and geographical location, and that this homophilic behavior presented variations and an intensification during the electoral period.

**Author Contributions:** Conceptualization, V.I.-T., J.L.M.-S.; methodology, V.I.-T. and J.L.M.-S.; software, J.L.M.-S.; validation, V.I.-T. and J.L.M.-S.; formal analysis, V.I.-T.; investigation, V.I.-T.; resources, V.I.-T., M.D.B. and J.L.M.-S.; data curation, V.I.-T.; writing—original draft preparation, V.I.-T.; writing—review and editing, J.L.M.-S.; visualization, V.I.-T.; supervision, J.L.M.-S.; project administration, J.L.M.-S. All authors have read and agreed to the published version of the manuscript.

**Funding:** This research was funded by the Spanish Ministry of Economy, Industry, and Competitiveness as part of the project "Influencers in Political Communication in Spain. Analysis of the Relationships Between Opinion Leaders 2.0, Media, Parties, Institutions, and Audiences in the Digital Environment (R + D + Project), grant number CSO2017-88620-PF.

**Institutional Review Board Statement:** Not applicable.

**Informed Consent Statement:** Not applicable.

**Data Availability Statement:** Not applicable.

**Conflicts of Interest:** The authors declare no conflict of interest. The funders had no role in the design of the study; in the collection, analyses, or interpretation of data; in the writing of the manuscript, or in the decision to publish the results.

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
