# Peer review of "Who Did Spanish Politicians Start Following on Twitter? Homophilic Tendencies among the Political Elite"

_socsci, doi:10.3390/socsci11070292_

Round 1
Reviewer 1 Report
This is an interesting study but it also raised a few questions:
My biggest concern is the sampling method and its description. Authors speak of a sample and a network but how the nodes of the network are defined/selected is not clear. They also tell that they used specific software for sampling their network but this software is a total black box for the reader. It is not clear what exactly it does and how it does it. As far as I understand, the authors selected Twitter accounts of (all?) sitting Spanish MPs. This is a sample of political elites in my view, not a network. They then analyzed on aggregate which accounts were followed recently by these politicians as a group over several years. This approach raises several questions: Is it methodologically valid to treat politicians as an aggregate? What is overlooked by this kind of approach?
My second concern is related to the first: As the theoretical grounding of the paper is homophily, I was wondering if it makes much sense to study this on an aggregate level. At least, this should be discussed. Are politicians so homogenous that they are as a group similar to other groups or other individuals and how can this be justified? If one chooses such a perspective, what can we learn from it about the power structure of society? Relatedly, I was really stunned that political ideology or party sympathy was completely left out as characteristic defining homophily. It would make much sense to assume that politicians interact with those who are politically close to their own views and goals. Other characteristics such as geographic origin are less convincing. especially as I would expect that Spanish MPs are culturally closer to actors from other southern EU countries (Italy, Portugal, France, Greece, perhaps Germany) that from Northern countries like Sweden or UK. US may be a different story due to the global influence of US politics but still, it's questionable why such contacts should take precedence over interactions with Latin America? These contradictions must be explained and discussed in relation to the concept of homophily. Only after a clarification of the concept and a better discussion of the results in light of the clarified theoretical framework are the findings really relevant.
The claim in the discussion (323-325) that findings support homophily as an explaining factor for interaction behavior is not supported by the results unless the points above are clarified. The finding concerning the tendency of politicians to follow/network with journalists is more convincingly explained in the discussion.
Author Response
Dear reviewer,
thank you very much for your time and input. We have taken into account your comments and suggestions and have made adjustments to the text in order to include them. Below we detail the changes made, which can be found in blue in the body of the text of the attached document.
My biggest concern is the sampling method and its description. Authors speak of a sample and a network but how the nodes of the network are defined/selected is not clear. They also tell that they used specific software for sampling their network but this software is a total black box for the reader. It is not clear what exactly it does and how it does it. As far as I understand, the authors selected Twitter accounts of (all?) sitting Spanish MPs. This is a sample of political elites in my view, not a network. They then analyzed on aggregate which accounts were followed recently by these politicians as a group over several years. This approach raises several questions: Is it methodologically valid to treat politicians as an aggregate? What is overlooked by this kind of approach?
My second concern is related to the first: As the theoretical grounding of the paper is homophily, I was wondering if it makes much sense to study this on an aggregate level. At least, this should be discussed. Are politicians so homogenous that they are as a group similar to other groups or other individuals and how can this be justified? If one chooses such a perspective, what can we learn from it about the power structure of society? Relatedly, I was really stunned that political ideology or party sympathy was completely left out as characteristic defining homophily. It would make much sense to assume that politicians interact with those who are politically close to their own views and goals.
We have expanded the explanation on the procedure carried out to determine and construct the sample. We have also provided an explanation on the software used and its functionalities, as well as theoretical justification for determining the network.
We also discussed why we aggregate the politicians as the unit of members of the Spanish Parliament, and comment on other possible criterion, with a justification of the selected one and how this impacts in the homophily analysis. Definitions of the political elite have been provided and a theoretical reflection on them has been added to justify their study. In addition, we explained that the sample covers all members of the Spanish parliament, seeking to analyze the Spanish political class as a unit, as all the deputies that coincided in the parliament from 2017 to 2020 were analyzed. They are heterogeneous in terms of party affiliation, gender, age, origin, among other variables, but are homogeneous in terms of the social role they occupied in the studied period, and therefore homophily can be measured in terms of similarity to the determined sample.
“Materials and Methods
With the aim of understanding the behavior of the Spanish politicians regarding who they started following on Twitter, we created a sample of deputies. This sample was composed by the deputies that coincided in the parliament during the studied period, which covered the years 2017 to 2020. To define the sample, we made a database with all deputies who integrated the parliament between 2017 and 2020 and then proceeded to select those who coincided during these four years. This means that all those deputies who were only during a shorter period within those years and not the whole period, were removed. This way, we were left with those who shared the four years of parliamentary duty.
We manually checked the number of followers, location and gender of the members of the sample and once we identified it, we proceeded to create a network, understood as such according to social network analysis (Barnes & Harary, 1983; Casero-Ripollés, 2021b; Grandjean, 2016; Tang & Liu, 2010), in order to analyze them. Utilizing a machine learning software named Contexto.io, which was developed as part of the project “Influencers in Political Communication in Spain. Analysis of the Relationships Between Opinion Leaders 2.0, Media, Parties, Institutions, and Audiences in the Digital Environment”. This software can organize, explore and analyze contexts of information around people using their public digital footprints. A context is composed by a group of people and/or organizations that interact forming an ecosystem. They are created by using their Twitter accounts which are then algorithmically sorted by their relevance within the context taking into account their digital trace. Therefore, we performed a manual search of each of the deputies on Twitter to identify their user accounts. Utilizing the above-mentioned software we created a new group and manually added each Twitter user and thus created the network with the 97 Twitter accounts of the deputies who coincided in the Spanish parliament between 2017 and 2020. Once the network is created, this software organizes the accounts in a graph regarding different possible parameters such as Relations, Communication, Common organizations and Predicted links, which is the categorizations we selected for present the sample.
(…)
Once we created the sample, we consulted the data regarding who they started to follow in different periods. The sample, composed by all the deputies that coincided in the Spanish Parliament from 2017 to 2020 is understood as one possible cut to define the stable political elite of those years, in order to have a sample with sufficient members to analyze as a conjunct. We could have categorized the sample in many ways, taking into account the politicians' gender, race, origin, political affiliation, religious affiliation, and analyze homophilic tendencies from these possible different categories (McPherson et al., 2001). The present study represents a specific case study on Spanish politicians on Twitter, so we decided to make an approximation to the homophilic behaviors of the whole political class that composed the Parliament during four years, making an approximation to the macro category as politicians in power, to see if they started to follow the citizenry or if they started to follow mainly other politicians and media, as stated in previous research on echo chambers and homophily on Twitter (Bruns & Highfield, 2013; Colleoni et al., 2014a).
Moreover, it is known that there is no consensus when referring to the concept of political elite (Zuckerman, 1977). Nonetheless, taking into account different definitions of the concept, such as an elite that has a preeminent political influence (Roberts, 1971); the Weberian model of elite power understood in terms of those who are in stable positions at the top of relevant social institutions (Wedel, 2017); the concept of the elites as those who are in the position to make decisions that impact other individuals´ lives by being in most relevant social hierarchies and institutions (Mills, 1956); or as the minority that rules the society (Rahman Khan, 2012). Moreover, elites can be understood under Meisel´s umbrella of the 3Cs, where there is group consciousness, coherence and conspiracy among the members of a power group (Korom & Planck, 2015; Meisel, 1958; Zuckerman, 1977). Therefore, in the present research we study the Spanish political elite from the perspective of a power group that exercises high influence and that can be analyzed as a cluster representing those who were in a hierarchical position in one of the most influential institutions, the Parliament, enabling them to make decisions that affect the rest of the members of the society, as they are all the deputies who integrated the parliament from 2017 to 2020, contemplating exclusively those who shared the entire period analyzed, with the purpose of generating a first approximation to their behavior regarding the type of accounts they began following. They are heterogeneous in terms of party affiliation, gender, age, origin, among other variables, but are homogeneous in terms of the social role they occupied in the studied period, and therefore homophily can be measured in terms of similarity to the determined sample. We believe there are lines to further explore in future research by subcategorizing this elite in different periods, by political party or by gender.
We were also able to access the data of the accounts they started following through the Contexto.io software, which has a section called Expand where it is possible to visualize the accounts that the context started to follow, with possibility of selecting specific periods to analyze. This section provides the option to select whether to display the accounts that the group started to follow including those belonging to the context or excluding them or to display only those that are outsiders of the network. The software thus provides a list in order of popularity within the network, measured by the percentage of users in the group that started following each account. For this study, we chose to visualize the accounts that the sample started to follow both, in-network and out-of-network.”
Other characteristics such as geographic origin are less convincing. especially as I would expect that Spanish MPs are culturally closer to actors from other southern EU countries (Italy, Portugal, France, Greece, perhaps Germany) that from Northern countries like Sweden or UK. US may be a different story due to the global influence of US politics but still, it's questionable why such contacts should take precedence over interactions with Latin America? These contradictions must be explained and discussed in relation to the concept of homophily. Only after a clarification of the concept and a better discussion of the results in light of the clarified theoretical framework are the findings really relevant.
With respect to the concern about homophily in relation to geographic location, what was pointed out about the global North and South is related to decolonialist theories. We agree it may not contribute in the present case, so we removed this reflection in order to focus on the dimension of geographic location as a generic category of analysis and widely used in studies on homophily.
The claim in the discussion (323-325) that findings support homophily as an explaining factor for interaction behavior is not supported by the results unless the points above are clarified. The finding concerning the tendency of politicians to follow/network with journalists is more convincingly explained in the discussion.
We added definitions of homophily that may help clarify why the fact that de politicians began following other politicians is considered a homophilic behavior.
Reviewer 2 Report
Thank you very much for giving me the chance to review this excellent piece researching the homophilic tendencies among the political elite.
The article is well organized, well written, and methodologically sound, thus it represents a necessary contribution to the stream of literature looking at whether Twitter actually contributes to a more democratic political debate or it functions as echo chamber of the elites.
However, I believe that this work presents some flaws that I am sure the authors will be able to easily fix.
1) LITERATURE REVIEW
Although the authors discuss previous literature about Twitter, but since the case is under scrutiny in Spain (and this Journal has no word limits) it would be helpful to provide an overview of what has been published about politics and Twitter in Spain.
Acknowledging this literature would help ground the study.
Some example:
Casero-Ripollés A. Influencers in the Political Conversation on Twitter: Identifying Digital Authority with Big Data. Sustainability. 2021; 13(5):2851. https://doi.org/10.3390/su13052851
Haman, M., & Školník, M. (2021). Politicians on Social Media. The online database of members of national parliaments on Twitter. Profesional De La Información, 30(2). https://doi.org/10.3145/epi.2021.mar.17
Cervi, Laura; Roca-Trenchs, Núria (2017). “Towards an Americanization of political campaigns? The use of Facebookand Twitter for campaigning in Spain, USA and Norway”. Anàlisi, n. 56, pp. 87-100. https://doi.org/10.5565/rev/analisi.3072
2) METHODS
LINE 126: With the aim of understanding the behavior of the Spanish politicians regarding who they started following on Twitter, we created a sample of deputies.
Author(s) should explain how the data set was created (criteria for inclusion and exclusion, etc.)
LINE 130: utilizing a machine learning software named Contexto, which was developed as part of the project…
Same. The software should be explained (creation, how it works, etc.)
3) GENDER PERSPECTIVE
LINE 178:
a) From the accounts that belonged to people we categorized them according to the gender they identify themselves with or by analyzing the profile (description and picture).
Since this is a very delicate issue I suggest to explain the categorization method in detail.
Probably it is just a matter of word choice: but, for instance, how can we be sure that a body culturally read as a “man” (in his picture) actually identifies with the masculine gender?
b) The Gender subcategories were Women, Non-binary and Men
Same. Please justify why you chose “non-binary” over other definitions. Quoting literature could be helpful to justify your choice.
Good luck!
Author Response
Dear reviewer,
Thank you very much for your time and suggestions. We have worked on each point proposed, with the conviction that they have contributed to improve the work. Below we comment on the work done on each point. You can also find the changes made in the text the color blue.
1) LITERATURE REVIEW
Although the authors discuss previous literature about Twitter, but since the case is under scrutiny in Spain (and this Journal has no word limits) it would be helpful to provide an overview of what has been published about politics and Twitter in Spain.
Acknowledging this literature would help ground the study.
Some example:
Casero-Ripollés A. Influencers in the Political Conversation on Twitter: Identifying Digital Authority with Big Data. Sustainability. 2021; 13(5):2851. https://doi.org/10.3390/su13052851
Haman, M., & Školník, M. (2021). Politicians on Social Media. The online database of members of national parliaments on Twitter. Profesional De La Información, 30(2). https://doi.org/10.3145/epi.2021.mar.17
Cervi, Laura; Roca-Trenchs, Núria (2017). “Towards an Americanization of political campaigns? The use of Facebookand Twitter for campaigning in Spain, USA and Norway”. Anàlisi, n. 56, pp. 87-100. https://doi.org/10.5565/rev/analisi.3072
We have added the suggested authors, as well as a few more, who collaborated to provide insight into studies on politicians on Twitter.
Therefore, we included the following bibliography and below you can see the new content:
Beltran; Javier, Gallego; Aina, Huidobro; Alba, Romero; Enrique, and Padró; Lluís. (2021). Male and female politicians on Twitter: A machine learning approach. European Journal of Political Research, 60(1), 239–251. https://doi.org/10.1111/1475-6765.12392
Casero-Ripollés; Andreu. (2021). Influencers in the Political Conversation on Twitter : Identifying Digital sustainability. Sustainability, 13(March). https://doi.org/10.3390/su13052851
Cervi; Laura, and Roca; Núria. (2017). Cap a l’americanització de les campanyes electorals? L’ús de Facebook i Twitter a Espanya, Estats Units i Noruega. Analisi, 56, 87–100. https://doi.org/10.5565/rev/analisi.3072
Coesemans; Roel, and De Cock; Barbara. (2017). Self-reference by politicians on Twitter: Strategies to adapt to 140 characters. Journal of Pragmatics, 116, 37–50. https://doi.org/10.1016/j.pragma.2016.12.005
Fernández-Rovira; Cristina, and Villegas-Simón; Isabel. (2019). Comparative study of feminist positioning on twitter by Spanish politicians. Analisi, 61, 77–92. https://doi.org/10.5565/rev/analisi.3234
Guerrero-Solé; Frederic, and Perales-García; Cristina. (2021). Bridging the Gap: How Gender Influences Spanish Politicians’ Activity on Twitter. In Journalism and Media (Vol. 2, Issue 3, pp. 469–483). https://doi.org/10.3390/journalmedia2030028
Jungherr; Andreas. (2016). Twitter use in election campaigns: A systematic literature review. Journal of Information Technology and Politics, 13(1), 72–91. https://doi.org/10.1080/19331681.2015.1132401
Stier; Sebastian, Bleier; Arnim, Lietz; Haiko, and Strohmaier; Markus. (2018). Election Campaigning on Social Media: Politicians, Audiences, and the Mediation of Political Communication on Facebook and Twitter. Political Communication, 35(1), 50–74. https://doi.org/10.1080/10584609.2017.1334728
Suau-Gomila; Guillem, Pont-Sorribes; Carles, and Pedraza-Jiménez; Rafael. (2020). Politicians or influencers? Twitter profiles of pablo iglesias and albert rivera in the spanish general elections of 20-d and 26-j. Communication and Society, 33(2), 209–225. https://doi.org/10.15581/003.33.2.209-225
“In Spain, Twitter research has focused on the identification of influential actors in the political conversation using big data to detect digital authority (Casero-Ripollés, 2021a), the use that Spanish political leaders do of the social platform analyzed from different perspectives such as in comparison to politicians from different political systems like the United Stated of America and Norway (Cervi & Roca, 2017), to detect the influence degree and the types of strategic communications tactics that the Spanish leaders use on Twitter, as well as analyzing the interconnection between the politicians Twitter and media profiles (Suau-Gomila et al., 2020), or regarding the linguistic strategies that politicians use in self-referencing (Coesemans & De Cock, 2017). Moreover, previous research on Twitter in Spain has focused on gender gaps among politicians, showing how there are still differences between the attention and amplification that women receive in the political Twitter sphere (Guerrero-Solé & Perales-García, 2021), the differences in the language use between men and women politicians (Beltran et al., 2021), as well as the differences between women and men politicians from different Spanish parties when tweeting about feminist issues (Fernández-Rovira & Villegas-Simón, 2019).
In this research we focus on analyzing the accounts that Spanish politicians began following, with the aim of contributing in the research of the use that political actors to of Twitter in Spain with a gender perspective, which even though has been previously explored (Beltran et al., 2021; Casero-Ripollés, 2021a; Cervi & Roca, 2017; Coesemans & De Cock, 2017; Fernández-Rovira & Villegas-Simón, 2019; Jungherr, 2016; Stier et al., 2018; Suau-Gomila et al., 2020), still lacks the consideration of homophily among Spanish political elites on Twitter. Moreover, research on following flows on Twitter in Spain among politicians is practically non-existent”.
2) METHODS
LINE 126: With the aim of understanding the behavior of the Spanish politicians regarding who they started following on Twitter, we created a sample of deputies.
Author(s) should explain how the data set was created (criteria for inclusion and exclusion, etc.)
LINE 130: utilizing a machine learning software named Contexto, which was developed as part of the project…
Same. The software should be explained (creation, how it works, etc.)
We have expanded the explanation on the procedure carried out to determine and construct the sample. We have also provided an explanation on the software used and its functionalities.
“Materials and Methods
With the aim of understanding the behavior of the Spanish politicians regarding who they started following on Twitter, we created a sample of deputies. This sample was composed by the deputies that coincided in the parliament during the studied period, which covered the years 2017 to 2020. To define the sample, we made a database with all deputies who integrated the parliament between 2017 and 2020 and then proceeded to select those who coincided during these four years. This means that all those deputies who were only during a shorter period within those years and not the whole period, were removed. This way, we were left with those who shared the four years of parliamentary duty.
We manually checked the number of followers, location and gender of the members of the sample and once we identified it, we proceeded to create a network, understood as such according to social network analysis (Barnes & Harary, 1983; Casero-Ripollés, 2021b; Grandjean, 2016; Tang & Liu, 2010), in order to analyze them. Utilizing a machine learning software named Contexto.io, which was developed as part of the project “Influencers in Political Communication in Spain. Analysis of the Relationships Between Opinion Leaders 2.0, Media, Parties, Institutions, and Audiences in the Digital Environment”. This software can organize, explore and analyze contexts of information around people using their public digital footprints. A context is composed by a group of people and/or organizations that interact forming an ecosystem. They are created by using their Twitter accounts which are then algorithmically sorted by their relevance within the context taking into account their digital trace. Therefore, we performed a manual search of each of the deputies on Twitter to identify their user accounts. Utilizing the above-mentioned software we created a new group and manually added each Twitter user and thus created the network with the 97 Twitter accounts of the deputies who coincided in the Spanish parliament between 2017 and 2020. Once the network is created, this software organizes the accounts in a graph regarding different possible parameters such as Relations, Communication, Common organizations and Predicted links, which is the categorizations we selected for present the sample.
(…)
Once we created the sample, we consulted the data regarding who they started to follow in different periods. The sample, composed by all the deputies that coincided in the Spanish Parliament from 2017 to 2020 is understood as one possible cut to define the stable political elite of those years, in order to have a sample with sufficient members to analyze as a conjunct. We could have categorized the sample in many ways, taking into account the politicians' gender, race, origin, political affiliation, religious affiliation, and analyze homophilic tendencies from these possible different categories (McPherson et al., 2001). The present study represents a specific case study on Spanish politicians on Twitter, so we decided to make an approximation to the homophilic behaviors of the whole political class that composed the Parliament during four years, making an approximation to the macro category as politicians in power, to see if they started to follow the citizenry or if they started to follow mainly other politicians and media, as stated in previous research on echo chambers and homophily on Twitter (Bruns & Highfield, 2013; Colleoni et al., 2014a).
Moreover, it is known that there is no consensus when referring to the concept of political elite (Zuckerman, 1977). Nonetheless, taking into account different definitions of the concept, such as an elite that has a preeminent political influence (Roberts, 1971); the Weberian model of elite power understood in terms of those who are in stable positions at the top of relevant social institutions (Wedel, 2017); the concept of the elites as those who are in the position to make decisions that impact other individuals´ lives by being in most relevant social hierarchies and institutions (Mills, 1956); or as the minority that rules the society (Rahman Khan, 2012). Moreover, elites can be understood under Meisel´s umbrella of the 3Cs, where there is group consciousness, coherence and conspiracy among the members of a power group (Korom & Planck, 2015; Meisel, 1958; Zuckerman, 1977). Therefore, in the present research we study the Spanish political elite from the perspective of a power group that exercises high influence and that can be analyzed as a cluster representing those who were in a hierarchical position in one of the most influential institutions, the Parliament, enabling them to make decisions that affect the rest of the members of the society, as they are all the deputies who integrated the parliament from 2017 to 2020, contemplating exclusively those who shared the entire period analyzed, with the purpose of generating a first approximation to their behavior regarding the type of accounts they began following. They are heterogeneous in terms of party affiliation, gender, age, origin, among other variables, but are homogeneous in terms of the social role they occupied in the studied period, and therefore homophily can be measured in terms of similarity to the determined sample. We believe there are lines to further explore in future research by subcategorizing this elite in different periods, by political party or by gender.
We were also able to access the data of the accounts they started following through the Contexto.io software, which has a section called Expand where it is possible to visualize the accounts that the context started to follow, with possibility of selecting specific periods to analyze. This section provides the option to select whether to display the accounts that the group started to follow including those belonging to the context or excluding them or to display only those that are outsiders of the network. The software thus provides a list in order of popularity within the network, measured by the percentage of users in the group that started following each account. For this study, we chose to visualize the accounts that the sample started to follow both, in-network and out-of-network.”
3) GENDER PERSPECTIVE
LINE 178:
- a) From the accounts that belonged to people we categorized them according to the gender they identify themselves with or by analyzing the profile (description and picture).
Since this is a very delicate issue I suggest to explain the categorization method in detail.
Probably it is just a matter of word choice: but, for instance, how can we be sure that a body culturally read as a “man” (in his picture) actually identifies with the masculine gender?
- b) The Gender subcategories were Women, Non-binary and Men
Same. Please justify why you chose “non-binary” over other definitions. Quoting literature could be helpful to justify your choice.
We have expanded the explanation on the categorization of the beads in relation to their gender, clarifying that we have looked for the way in which the beads define themselves and we have added bibliography where the "non-binary" categorization can be referenced.
“From the accounts that belonged to people we categorized them according to the gender they identify themselves with or by analyzing the profile (description and picture). To do this, we took into account how they described themselves in their bios and if their bios didn't make it clear, we looked for more information online about each user to find out how they defined themselves. Since most of them use Spanish and Catalan, which are languages that contain gender differentiation in most of the words, it was easier to identify how they call themselves, since by putting for example “deputy” in their bios, which would be "diputada" or "diputado" or "diputade" in Spanish, we can already know how they identify gender-wise, as “a” is used for women, “o” for men and “e” for non-binaries. Another example is an account who´s bio was “Un socialista vasco”, which translates as “A basque socialist”. This phrase in Spanish clarifies the gender the user identifies with, as the pronoun is masculine. The Gender subcategories were Women, Non-binary and Men (Butler, 1988; Richards et al., 2016), aiming to explore gender balance (or dis-balance) trends, as women and dissidences have a long-lasting tradition of being underrepresented in power positions (Carli & Eagly, 2002; Connell, 1987; Kubu, 2017; Madsen & Andrade, 2018; Painter-Morland, 2011). Previous research has shown a problematic confusion between sex and gender, which tend to be presented as interchangeable categories, when sex has been defined as a biological phenomenon whereas gender is understood as a cultural dimension (Bittner & Goodyear-Grant, 2017). Both, sex and gender, tend to be understood as binary categories, male and female in the case of sex, and men and women in the case of gender, whereas research has proven that both are not. There is a percentage of the population that is born as intersex or third-sex (Carpenter, 2018), estimated to be around the 1,7% (Amnesty, 2018), and as there are other gender identities such as genderqueer and non-binary (Richards et al., 2016). In this study, following previous research where identities who do not identify themselves in a binary way as women or men are taken into account, we categorized the accounts in Women, Men and Non-binary (Medeiros et al., 2020).”
Round 2
Reviewer 1 Report
The paper has improved in clarity and it cites an impressive amount of relevant literature thereby giving a good, initiual overview of political communication in the social media ecosystem. The general idea to study homophily among MPs is also relevant and convincing. However, the weakness remains a superficial appraoch by treating MPs as a homogenous group without properly arguing why this apporach is fruitful despite obvious caveats.
The theoretical argument that would explain which common trait breeds which following behavior among the MPs is still underdeveloped. Fragements are now presented in lines 181-216, but it should be further elaborated and moved to the section before where the relevant literature is discussed.
As already stated in the first review, geography as a common trait is a misleading term here: What is meant is a political-cultural proximity of politicians of affluent, western democracies.
The RQ about gender should be dropped as it cannot be studied with the approach taken. Author(s) would have needed to subdivide their sample into two groups to look into gendered afffinity.
It is odd taht party/ideology is not tested. Why wasn't it possible to break down the sample by party to look for ideoligical homophily? Lumping together all MPs glosses over this important differences in their political mind. At the very least, this sould be discussed as a clear limitation.
There are also problems with presentation of results, especially percentages and colors in Figures: Figure 2 too small, illisibile; Figure 3 shows no political accounts in 2019 (??); Figure 6: How can percentages for men and women be either more than or below 100 per cent in the categories??
Please clarify the statement in l. 415 by adding "as a group" behind deputies.
Author Response
The paper has improved in clarity and it cites an impressive amount of relevant literature thereby giving a good, initiual overview of political communication in the social media ecosystem. The general idea to study homophily among MPs is also relevant and convincing. However, the weakness remains a superficial appraoch by treating MPs as a homogenous group without properly arguing why this apporach is fruitful despite obvious caveats. The theoretical argument that would explain which common trait breeds which following behavior among the MPs is still underdeveloped. Fragements are now presented in lines 181-216, but it should be further elaborated and moved to the section before where the relevant literature is discussed.
We have moved part of this section to the section on the theoretical framework related to homophily, leaving the first lines more related to methodology in the methodology section.
In the methodological section we also added:
Methodologically, in elite studies, there are three main ways of determining an elite for its study: positional, decisional and reputational (Best et al., 2017; Hoffmann‐Lange, 1989), also categorized as reputational, structural, and the agency or decision-making approach (Scott, 1974). In the present study, we have taken the positional/structural path, since, as Scott states: “the structural approach has the most to offer to researchers on power and that it provides a basis for incorporating the insights of the rival approaches” (Scott, 1974) p.84. Taking into account theoretical and pragmatic reasons, the positional method is one of the most widely used in the study of national elites (Best et al., 2017; Hoffmann‐Lange, 1989; Larsen & Ellersgaard, 2017) p.53. Given that the present study is a first approach to the political homophilic tendencies regarding the accounts that the Spanish political elite began following, we believe that the best methodological approach is to select the sample according to its formal position of power in society, in this case the set of deputies that conform the Spanish parliament. Structural approaches to power are centered on the aspects of strategic positions in the main institutions of a society. Positions that are the at the core of the resource’s distribution and control, which are the main centers of power, and therefore, those who occupy these positions are understood as main actors in the exercise of power. Therefore, the sample represents an elite with a clear cut that seeks to provide an approximation of the political elite in Spain. Like any method and methodological decision, it has advantages and disadvantages. The advantage in this case is to be able to understand how the Spanish elite operates as a whole, as a group of decision-makers, as a cluster of people with positions of high impact on citizens lives. The limitation of this approach is to leave aside the differences among them, such as gender, political orientation, nationality, language they speak. We believe it would be interesting to deepen into the abovementioned subcategories in future research, subsequently to the present one that aims to analyze the parliamentary Spanish elite as a group, as even they are heterogenous, the political elite´s diversity has been presented by authors as more apparent than real, taking into account that they share the involvement in central policy decisions (Best et al., 2017). Moreover, we follow the methodological approach of several previous studies where the political elite is analyzed as such, leaving aside the differences among them, such as their political affiliation or gender (D’heer & Verdegem, 2014; Putnam, 1976; Sjöberg & Drottz-Sjöberg, 2008; Verweij, 2012).
In the theoretical framework we added:
There is no consensus when referring to the concept of political elite (Zuckerman, 1977). Nonetheless, taking into account different definitions of the concept, such as an elite that has a preeminent political influence (Roberts, 1971); the Weberian model of elite power understood in terms of those who are in stable positions at the top of relevant social institutions (Wedel, 2017); the concept of the elites as those who are in the position to make decisions that impact other individuals´ lives by being in most relevant social hierarchies and institutions (Mills, 1956); or as the minority that rules the society (Rahman Khan, 2012). Moreover, elites can be understood under Meisel´s umbrella of the 3Cs, where there is group consciousness, coherence and conspiracy among the members of a power group (Korom & Planck, 2015; Meisel, 1958; Zuckerman, 1977). Therefore, in the present research we study the Spanish political elite from the perspective of a power group that exercises high influence and that can be analyzed as a cluster representing those who were in a hierarchical position in one of the most influential institutions, the Parliament, enabling them to make decisions that affect the rest of the members of the society, as they are all the deputies who integrated the parliament from 2017 to 2020, contemplating exclusively those who shared the entire period analyzed, with the purpose of generating a first approximation to their behavior regarding the type of accounts they began following. They are heterogeneous in terms of party affiliation, gender, age, origin, among other variables, but are homogeneous in terms of the social role they occupied in the studied period, and therefore homophily can be measured in terms of similarity to the determined sample. We believe there are lines to further explore in future research by subcategorizing this elite in different periods, by political party or by gender. In the present research we study the Spanish political elite as a group, taking into account the positional method of elite studies (Best et al., 2017; Hoffmann‐Lange, 1989) that states that political power and influence in societies is conferred by formal institutional positions in the main organizations where decisions that affect the citizenship are taken, as well as the institutions responsible for the resources social distribution (Best et al., 2017). The elite structure is pluralistic, nonetheless “theorists acknowledge that modern democracies are organizationally diverse, they claim that the diversity of organizations and interests they embody are not reflected in the elite structure. They assume that power is more concentrated in a small power elite than exponents of pluralism believe, so that participation in crucial policy decisions is limited to a small circle or knot of actors with common social backgrounds and interests that are concealed by a diversity of organizations and interests that, in terms of decisive power, is more apparent than real” (Best et al., 2017) p.80.
As already stated in the first review, geography as a common trait is a misleading term here: What is meant is a political-cultural proximity of politicians of affluent, western democracies.
Geography is a variable defined in several investigations to measure homophily. Indeed we sought to know the homophily in the terms of geographical location in order to understand if as well as it has been analyzed in previous studies (Ausserhofer & Maireder, 2013; Casero-Ripollés, 2021), it is something that could be observed in the case of the Spanish political elite, and indeed we found that it is, so we can add knowledge in this area, finding results that demonstrate that the Spanish political elite follow a large majority of accounts from their same country.
The RQ about gender should be dropped as it cannot be studied with the approach taken. Author(s) would have needed to subdivide their sample into two groups to look into gendered afffinity.
In this study, we seek to understand the space of women in the political elite in Spain. In future research it may be interesting to separate the sample by gender. Given that previous studies have demonstrated that even though gender representation is an important factor, having at least an equal number of women representatives in government does not necessarily mean that there will be a better qualitative representation of women interests (Lombardo, 2008). Women in elite environments may give more space to men, as they are traditionally associated with power (Connell, 1987). Therefore, in this research we are interested in understanding the behavior of the Spanish elite by analyzing the members of the parliament in relation to gender.
We added the following text in the theoretical frame:
Moreover, even when having balanced gender representation, an equal number of women representatives in the government does not necessarily mean that there will be a qualitative representation of women interests (Lombardo, 2008). Regarding social media interactions, it has been stated in previous research how men journalists and politicians tend to interact with a majority of male peers (Colleoni et al., 2014a; Usher, 2018), whereas such inbreeding homophily has not been found among women journalists (Maares et al., 2021).
It is odd taht party/ideology is not tested. Why wasn't it possible to break down the sample by party to look for ideoligical homophily? Lumping together all MPs glosses over this important differences in their political mind. At the very least, this sould be discussed as a clear limitation.
We added the limitation in several sections:
Like any method and methodological decision, it has advantages and disadvantages. The advantage in this case is to be able to understand how the Spanish elite operates as a whole, as a group of decision-makers, as a cluster of people with positions of high impact on citizens lives. The limitation of this approach is to leave aside the differences among them, such as gender, political orientation, nationality, language they speak.
The analysis of political ideology is a limitation of the present research, in which we decided to focus on the types of accounts, number of followers, geographic location, and gender. We consider it is relevant to delve into more variables of analysis in future research, such as political ideology.
There are also problems with presentation of results, especially percentages and colors in Figures: Figure 2 too small, illisibile;
We have redesigned this graphic in order to present it with a larger font number.
Please find the new image in Figure 2.
Figure 3 shows no political accounts in 2019 (??)
There are three categories within political accounts, as it is explained in the methodology and in the references of the graph: public institutions, political parties and politicians.
In the year 2019 they only followed one type of account within the political accounts which were the accounts belonging to politicians, so in the year 2019 we can see 100% of accounts of politicians. Since that year they did not start following neither political parties nor public institutions, there are no percentages present in the graph of those types of accounts that year.
Figure 6: How can percentages for men and women be either more than or below 100 per cent in the categories??
This graph represents the categories from the accounts they started following divided by gender. The light blue columns represent the accounts that the sample started following that belong to women, divided between politicians (69%), journalists (17%) and users (14%). If we add up the percentages of these columns, we get 100, which represents the total percentage of women they started following.
The blue columns represent the men accounts that they started following, divided between politicians (57%), journalists (16%) and users (27%). Adding the percentages of these columns also gives 100, which represents the total percentage of men they started following.
We offer an alternative way of presenting the data in a new graph in case it is preferred, please find below the one presented before.
Please clarify the statement in l. 415 by adding "as a group" behind deputies.
Line 415 has the following text:
“ones. May it reflect a tendency to follow women only when they have a very clear establ”
We do not see the word deputies in this sentence.
We do see the word deputies in line 418 where it says” In this research we analyzed the accounts that the deputies that coincided in the”. We ask if this was the line you were referring to. If positive, we would add the suggested phrase “as a group” at the end of the sentence and not after deputies, given that after deputies we explain which ones, so the suggested phrase would go at the end of the sentence to maintain the meaning of the sentence.
Reviewer 2 Report
The authors have tackled all the issues and the paper is now publishable.
Author Response
Thank you once again for the comments and suggestions that helped improve this paper.
Round 3
Reviewer 1 Report
I believe the manuscript has now been sufficiently improved to warrant publication.